# Isolation of Antagonistic Endophytic Fungi from Postharvest Chestnuts and Their Biocontrol on Host Fungal Pathogens

**DOI:** 10.3390/jof10080573

**Published:** 2024-08-14

**Authors:** Yunmin Wen, Meng Li, Shuzhen Yang, Litao Peng, Gang Fan, Huilin Kang

**Affiliations:** 1College Key Laboratory of Environment Correlative Dietology, Ministry of Education, Wuhan 430070, China17853463773@163.com (M.L.);; 2College of Food Science and Technology, Huazhong Agricultural University, Wuhan 430070, China

**Keywords:** chestnut, antagonistic endophytic fungus, *N. parvum*, biocontrol, postharvest disease

## Abstract

In this study, antagonistic endophytic fungi were isolated from postharvest chestnut fruits; endophytic antagonistic fungi and their combination of inhibitory effects on the fungal pathogen *Neofusicoccum parvum* were evaluated. A total of 612 endophytic fungi were isolated from 300 healthy chestnut kernels, and 6 strains out of them including NS-3, NS-11, NS-38, NS-43, NS-56, and NS-58 were confirmed as antagonistic endophytic fungi against *Neofusicoccum parvum*; these were separately identified as *Penicillium chermesinum*, *Penicillium italicum*, *Penicillium decaturense*, *Penicillium oxalicum, Talarmyces siamensis*, and *Penicillium guanacastense*. Some mixed antagonistic endophytic fungi, such as NS-3-38, NS-11-38, NS-43-56, and NS-56-58-38, exhibited a much stronger antifungal activity against *N. parvum* than that applied individually. Among them, the mixture of NS-3-38 showed the highest antifungal activity, and the inhibition rate was up to 86.67%. The fermentation broth of NS-3, NS-38, and their combinations exhibited an obvious antifungal activity against *N. parvum*, and the ethyl acetate phase extract of NS-3-38 had the strongest antifungal activity, for which the inhibitory rate was up to 90.19%. The NS-3-38 fermentation broth combined with a chitosan coating significantly reduced *N. parvum* incidence in chestnuts from 100% to 19%. Furthermore, the fruit decay and weight loss of chestnuts during storage were significantly decreased by the NS-3-38 fermentation broth mixture along with a chitosan coating. Therefore, a mixture of *P. chermesinum* and *P. decaturense* could be used as a potential complex biocontrol agent to control postharvest fruit decay in chestnuts.

## 1. Introduction

The chestnut (*Castanea mollissima*) is an important economical crop in Asia, Europe, and America because of its delicious flavor and nutritional characteristics [1]. One of the major commercial difficulties is the high perishability of the product, which is mainly caused by pathogenic fungi [2,3]. The application of fungicides is the main method for controlling the postharvest disease of chestnut [4]. Although chemical fungicides are effective in controlling postharvest pathogens, the overuse of them has caused a series of problems, such as being a risk to the environment and consumer health, and the development of fungicide-resistance pathogens [5]. Due to this, safer and more environmentally friendly approaches to control postharvest diseases are urgently needed.

In recent years, there has been growing interest in using biological control agents (BCAs) as an alternative strategy, with the potential to eliminate the adverse effects of chemical control [6,7,8]. Endophytes are microorganisms that inhabit and colonize the internal plant tissue without causing visible damage or illness in the host [9,10]. There are many endophytes that have been tested as BCAs for controlling postharvest diseases [11,12,13]. For example, the endophytic fungi strain *Metschnikowia citriensis* could significantly inhibit the spore germination and mycelial growth of *G. citriaurantii* and effectively control the development of the sour rot of citrus fruits [14]. The endophytic fungi *Albifimbria verrucaria* from the leaves of wild grapes was demonstrated to have a wide range of biocontrol activity against grape pathogens, and its metabolites could significantly inhibit the mycelial growth and conidium germination of *B. cinerea* [15]. The endophytic fungus *Albifimbria verrucaria* from wild grape was an antagonist of *Botrytis cinerea* and other grape pathogens. The endophytic *Albifimbria pullulans S-2* can lead to changes in the bacterial and fungal community, inhibit decay incidence, maintain fruit firmness, and reduce the weight loss of tomatoes [16]. Furthermore, it has been demonstrated that endophytic fungi use at least three strategies to biocontrol the pathogens—the generation of antifungal substances, morphological alterations, and competition for nutrients and space. Therefore, using fungal endophytes as microbial biological control agents will offer an efficient and eco-friendly way to control postharvest diseases.

Our preliminary study found that postharvest chestnuts are rich in fungal species, including pathogenic fungi, beneficial endophytic fungi, and fungi that have no effect on the fruits [9]. However, studies on the fungi of postharvest chestnuts mainly focus on the isolation and control of pathogenic fungi [17,18]. Pathogenic fungi including *Neofusicoccum parvum*, *Cryphonectria parasitica*, *Gnomoniopsis smithogilvyi*, *Diplodina castanea*, *Gnomoniopsis castaneae*, *Penicillium expansum*, and *Penicillium griseofulvum* have been isolated from chestnuts [3,19,20,21,22,23,24,25,26,27,28]. However, studies on endophytic fungi of postharvest chestnuts are still limited. Therefore, the main aims of the current study were (i) to isolate and identify endophytic antagonistic fungi from postharvest chestnuts; (ii) to evaluate their antifungal activity against *Neofusicoccum parvum* from chestnuts; and (iii) to evaluate their biocontrol efficiency on the fruit decay of postharvest chestnuts.

## 2. Materials and Methods

### 2.1. Fruits and Pathogens

Chestnut fruits cv ‘Dahongpao’ were separately harvested at the mature stage from Luotian county, Hubei province, China. All samples were uniform in size and free of defects. The collected fruits were stored at 4 °C and were prepared for the experiment.

*N. parvum* was isolated from decayed chestnut fruits with typical symptoms, and was then confirmed based on the morphological characteristics of the colonies and DNA amplification of the internal transcribed spacer (ITS) region. The strains were cultured on potato dextrose agar (PDA) media at 26 °C and were stored at 4 °C.

### 2.2. Isolation of Endophytic Fungi

Endophytic fungi were isolated using the tissue separation method [29]. Briefly, the collected fruits were washed in running water and were then sterilized with 2% sodium hypochlorite (NaClO) and 75% ethanol for 2 min (three times). The samples were rinsed in sterile distilled water four times; then, the chestnut kernel was cut into small pieces (2 mm~5 mm) using a sterile scalpel. The kernel pieces were placed on PDA plates with cefotaxime (200 mg L^−1^) and were incubated at 26 °C for 4 days. The cultured fungi were considered endophytic fungi and were sub-cultured on individual PDA plates to obtain pure isolates for further identification.

### 2.3. Identification of the Fungal Isolates

The identification of endophytic fungi was performed using morphological and molecular analyses [30,31,32]. For the morphological analysis, the pure culture isolates were incubated at 26 °C. Colonies grown on each medium were distinguished on the basis of their appearance characteristics such as texture, color, sporulation, and diameters. The molecular analysis was carried out using the internal transcribed spacer (ITS1, ITS4, and BenA) sequences method [31,33]. Briefly, total DNA was extracted from the mycelium with a DNA extraction kit (Axygen, Silicon Valley, CA, USA) according to the manufacturer’s instructions. The primers specific for ITS, CaM, and BenA are presented in Table 1. Polymerase chain reaction (PCR) amplification of the extracted DNA was performed on a thermo cycler (Longgene Instruments Hangzhou, Hangzhou, China). The amplification conditions were as follows: initial denaturation at 94 °C for 3 min, followed by 24 cycles of 94 °C for 30 s, 54 °C for 30 s, 72 °C for 90 s, and a final extension at 72 °C for 10 min. After complete amplification, the PCR products were analyzed for gel electrophoresis using a 1% agarose gel stained with ethidium bromide solution (0.3 mg mL^−1^) and they were visualized under UV light. DNA sequencing was performed according to standard protocols using the custom sequencing services of Huayu Gene (Wuhan) Co., Ltd. (Wuhan, China). The DNA sequences were aligned with strain sequences downloaded from NCBI [34]. After manual correction, phylogenetic trees were constructed using the adjacency method using software MEGA 7.0 [35].

### 2.4. Determination of the Pathogenicity of Endophytic Fungi

The healthy chestnut fruits were rinsed in running water and were triple sterilized with 2% sodium hypochlorite (NaClO) for 2 min and were then washed with sterile distilled water. After air drying, the two wounds were made symmetrically at the waist and were inoculated with a 4-day-old mycelial disk (∅ 5 mm) of endophytic fungus, and a sterile PDA disk (∅ 5 mm) was used as the control. Then, the inoculated fruits were kept at 26 °C for 5 d. The lesion diameters were expressed as the mean of the width and length of the areas of decay.

### 2.5. Evaluation of Antagonistic Activity of Endophytic Fungi against N. parvum

The antagonistic activity of endophytic fungi against *N. parvum* was preliminarily determined using the dish confrontation method [36]. Briefly, a mycelial disk (∅ 5 mm) of 24-hour-old pathogens was placed in the center of the PDA plate before inoculating four mycelial disks (∅ 5 mm) of endophytic fungi around the fungal pathogen at a 30 mm distance. The confronted culture plates were incubated at 26 °C. Individual cultures of the pathogen were used as a control plate. After 7 days, the colony diameters were measured and the mycelia inhibition rate was calculated using the following equation:(1)$$mycelia\;inhibition\;rate=\frac{\left(Dc-Dt\right)}{\left(Dc-5\right)}\times 100\%$$
where D_c_ is the mean pathogenic fungus colony diameter (mm) of the control sets, and D_t_ is the mean pathogenic fungus colony diameter (mm) of the treatment sets.

### 2.6. Determination of Antifungal Activity of Three Organic Phase Extracts of Endophytic and Endophytic Fungal Fermentation Broths

The endophytic fungi NS-3, NS-38, and combinations of NS-3-38 were incubated in potato dextrose broth (PDB) and were cultured on a gyratory shaker at 26 °C for 7 days under 120 r/min conditions. Initially, the fermentation broth obtained was filtered through a microporous filter membrane of 0.45 mm to obtain sterile fermentation broth, and was then incubated at 26 °C. After 7 days, the colony diameters were measured and the mycelia inhibition rate was calculated.

The fungal culture filtrate was concentrated using rotary evaporation, and the concentrated solution was then fractionated using liquid–liquid extraction technique successively with petroleum ether, ethyl acetate, and *n*-butanol to obtain three fractions, i.e., a petroleum ether fraction (PE), an ethyl acetate fraction (EA), and an n-butanol fraction (BE).

The antifungal activity of PE, EA, and BE against *N. parvum* was analyzed according to a previous method with slight modifications [18]. In brief, 19 mL of sterilized PDA medium was mixed with 1 mL of the appropriate concentration of PE, EA, and BE (40% aqueous acetone as solvent) to achieve the final concentrations of 5 mg/mL PE, EA, and BE; they were then poured into Petri dishes (90 mm in diameter). A pathogenic fungus mycelial disk (5 mm in diameter) from 4-day-old fungal cultures was placed in the center of each Petri dish and was then incubated at 26 °C. After 7 days, the colony diameters were measured and the mycelia inhibition rate was calculated using the following equation:(2)$$mycelia\;inhibition\;rate=\frac{\left(D0-Dt\right)}{\left(D0-5\right)}\times 100\%$$
where D_0_ is the mean colony diameter (mm) of the control sets, and D_t_ is the mean colony diameter (mm) of the treatment sets.

### 2.7. Effect of Culture Filtrate from Antagonistic Endophytic Fungi on Disease Severity of Chestnuts Caused by N. parvum

The chestnuts were wounded (5 mm width × 5mm length) symmetrically at the equatorial side and were then separately coated with chitosan solution dissolved with culture filtrate from antagonistic endophytic fungi or distilled water. And then, a 4-day-old fungal disk of *N. parvum* was inoculated and placed at 26 °C for 5 d. The fruits treated only with water were used as the controls. The lesion diameter was measured and the disease severity was defined as the size of lesion diameter (mm), as follows: scale 0, 0 mm (no decay); scale 1, 1–5 mm; scale 2, 5–10 mm; scale 3, 10–20 mm; and scale 4, >20 mm. The disease severity was calculated according to the formula below:(3)$$disease\;severity=\frac{\sum disease\;scale\times number\;of\;fruit\;in\;each\;scale}{highest\;disease\;scale\times number\;of\;total\;fruit}\times 100\%$$

### 2.8. The Effect of Antagonistic Endophytic Fungi on Chestnut Fruit Decay during Storage

The fruits were soaked in chitosan solution dissolved with fermentation broth of antagonistic endophytic fungi for 5 min. The fruits treated with chitosan solution were used as the controls. After air drying, the fruits were stored at 25 ± 1 °C for 16 days. The weight loss and decay incidence of fruit during storage were separately measured according to the following Formulas (4) and (5), respectively.
(4)$$weight\;loss=(m_{1}-m_{0})/m_{1}\times 100\%$$
where m_1_ is the final weight of chestnuts, and m_0_ is the initial weight of chestnuts.
(5)$$decay\;incidence=M_{1}/M_{0}\times 100\%$$
where M_1_ is the number of moldy chestnuts, and M_0_ is the total number of chestnuts.

### 2.9. Statistical Analysis

All experiments employed a completely randomized design. Each experimental procedure was conducted in triplicate. All data were expressed as means ± standard deviation. Data were analyzed using one-way analysis (ANOVA) using SPSS 25.0. Duncan’s multiple range test was used to identify significant differences at *p* < 0.05.

## 3. Results

### 3.1. Isolation of Non-Pathogenetic Endophytic Fungi from Postharvest Chestnuts

In this study, a total of 612 endophytic fungi were isolated from 300 healthy chestnut kernels of “Dahongpao” chestnut from Luotian county. According to the observation of colony morphology, the isolated endophytic fungi could be roughly divided into 58 categories numbered NS-1-NS-58 in sequence. As shown in Figure 1, six endophytic fungi including NS-3, NS-11, NS-38, NS-43, NS-56, and NS-58 were confirmed to be non-pathogenetic to chestnuts; these were further used for evaluating their antagonistic activity against *N. parvum*.

### 3.2. Antagonistic Activity of the Non-Pathogenetic Endophytic Fungi against N. parvum

The results showed all of the six endophytic fungi appeared to demonstrate antagonistic activities against *N. parvum* (Table 2), and the inhibition rates of strains NS-3, NS-11, NS-38, NS-43, NS-56, and NS-58 against *N. parvum* were 51.85 ± 1.39%, 69.26 ± 1.72%, 44.07 ± 1.14%, 59.26 ± 1.59%, 45.19 ± 4.12%, and 42.22 ± 0.45%, respectively.

The antagonistic effects of six endophytic fungi against the pathogen were further determined. It was found that the combinations of NS-3-38, NS-11-43, NS-11-56, NS-38-43, NS-38-56, NS-38-58, NS-43-56, NS-43-58, NS-56-58-38, and NS-56-58-43 had a higher antagonistic activity, for which inhibition rates reached above 70%. Among them, the combination of NS-3-38 had the highest inhibition rates, at 86.67 ± 0.45%, 75.93 ± 3.86%, 76.48 ± 2.10%, 74.63 ± 1.39%, 76.67 ± 1.98%, 75.93 ± 0.26%, 75.00 ± 2.36%, 77.22 ± 1.64%, 76.11 ± 0.45%, 77.59 ± 2.58%, and 72.41 ± 2.05%. The combination of endophytic fungi NS-3-38 had the most significant antifungal effect against the pathogen. However, not all combinations of strains showed a decrease in inhibition rate against the pathogen, such as the combination of NS-3-11, NS-3-43, NS-3-38-43, NS-56-58-11, and NS-F. This may be due to the increased competition between the fungi, which is as a result of increased nutrient depletion in mixed cultures, or the presence of growth-inhibiting substances in the metabolites of the mixed strains.

### 3.3. The Identification of Antagonistic Endophytic Fungi

The identification of antagonistic endophytic strains was performed using morphological and molecular analyses. From Figure 2, the colonies of all tested strains on PDA are circular with white margins. The colony color of NS-3 is light gray–green, and the fungal colonies of NS-38 have a dark grayish-green color. Microscopically, both have conidiophore in a typical whorled pattern (Figure 2A,C). The colonies of NS-11 were pale grayish-green, abaxially pale tea-brown, and conidiophores were fasciculated (Figure 2B). The colonies of NS-43 were dark green, wide-spreading, rounded, tomentose, and abaxially yellowish to yellow, with broom-like branches that are both irregular and closely spaced (Figure 2D). The colonies of NS-56 were yellowish-green to dark green, their surface was roughly rounded, and they were velvety crusted, abaxially colorless, and had closely arranged conidiophores (Figure 2E). The colonies of NS-58 were yellow–green, they had a rough surface, irregular edges, and a yellow abaxial surface, with irregular and highly branched mesophyll branches (Figure 2F). The morphology of these tested fungus were closely related to those of *Penicillium* sp., as described in previous studies.

As shown in Figure 3A, the sequencing reads of NS-3 based on the molecular analysis of the ITS and BenA regions were aligned with those of the annotated *Penicillium chermesinum* strain DTO 298-I8 in the NCBI database with 100% identity. The phylogenic tree constructed using the NJ method based on ITS and BenA gene sequences indicated the strain NS-3 was placed in the same clade as *P. chermesinum*. Therefore, the strain NS-3 was identified as *P. chermesinum*. The nucleotide sequences of BenA and CaM of strain NS-38 were 77.31% and 83.48%, which is identical to *Penicillium decaturense* strain CBS 117509, thus enabling the identification of NS-38 as *P. decaturense* (Figure 3C). From Figure 3B,D, the cluster analysis of the ITS sequence showed that the strain NS-11 had the closest relationship with *Penicillium italicum* (97.26% and 97.83% sequence similarity) and the strain NS-43 had the closest relationship with *Penicillium Oxalicum* (99% sequence similarity). According to their ITS sequence analysis, the BenA and CaM nucleotide sequences of strain NS-56 matched with *Talaromyces siamensis* strain CBS 475.88 with a matching degree of 98.9% and 99.8%; therefore, NS-56 was identified as *T. siamensis* (Figure 3E). According to Figure 3F, the BenA and CaM nucleotide sequences of strain NS-58 were 100% and 99.44%, identical to *Penicillium guanacastense* strain AS3.15361, respectively, thus identifying NS-58 as *P. guanacastense*.

### 3.4. Antifungal Activity of Fungal Culture Filtrate Extracts from NS-3 and NS-38 against N. parvum

As shown in Figure 4A,B, the fungal culture filtrates of NS-3, NS-38, and NS-3-38 exhibited a significant antifungal activity against *N. parvum*. Among them, NS-3-38 had the highest inhibitory rate, which was up to 89.44 ± 1.45%. The antifungal activities of fungal culture filtrate from NS-3, NS-38, and NS-3-38 fractionated with different solvents were further evaluated (Figure 4C–H). The results showed all of the fractions appeared to demonstrate obvious inhibitory effects on the pathogens. Among them, the fractions extracted with ethyl acetate had stronger activities than those extracted with petroleum ether or 1-butanol. The fraction extracted with ethyl acetate from NS-3-38 had the highest inhibitory rate, reaching up to 90.19 ± 0.26%.

### 3.5. Effects of Culture Filtrate from NS-3, NS-38, and NS-3-38 on Disease Severity of Chestnuts Caused by N. parvum

In this study, the inhibitory effects of NS-3, NS-38, and NS-3-38 combined with chitosan on the fruit decay of chestnut fruits caused by *N. parvum* were evaluated, and the results are shown in Figure 5. The culture filtrates from NS-3, NS-38, or NS-3-38 combined with chitosan appeared to be an effective control on fruit decay caused by *N. parvum* (Figure 5A)*,* whose disease indexes were only 28.33% ± 1.06%, 32.30% ± 1.59%, and 19.00% ± 1.64%, respectively. However, the disease indexes of the fruits in the control group and the chitosan coating group were 100% and 69.18 ± 2.49%, which were much higher than those of the treatment groups (Figure 5B). The results indicated NS-3, NS-38, and NS-3-38 fermentation broth along with chitosan coating could significantly inhibit the fruit decay caused by *N. parvum*, especially for NS-3-38.

### 3.6. Effects of the Culture Filtrate from NS-3, NS-38, and NS-3-38 on Fruit Decay and Weight Loss of Chestnuts during Storage

The fruit decay and weight loss of chestnuts coated with fermentation broth from NS-3, NS-38, and NS-3-38 combined with chitosan were evaluated during storage. From Figure 6A, it can be seen that fruit decay incidences in the treatment groups were obviously higher than those of chestnuts fruits coated with chitosan or those in the control group. Among them, the culture filtrate from NS-3-38 combined with chitosan had the strongest control effects on the fruit decay of chestnuts, whose incidence was only 20.00 ± 2.72%, which is much lower than those in the control group or the group coated only with chitosan. The weight loss of chestnut fruits treated with fermentation broth from NS-3, NS-38, and NS-3-38 combined chitosan exhibited a much lower weight loss when compared with the fruits in the control group or the group only treated with chitosan. The fruits treated with culture filtrate from NS-3-38 combined with chitosan had the lowest weight loss, which was only 45.5% of control fruits (Figure 6B). Therefore, the culture filtrate from NS-3-38 along with chitosan could control the fruit decay and prevent the chestnuts from losing weight during storage.

## 4. Discussion

Compared with other microbial strains as BCAs, the endophytic microbiome can more easily be administered, to penetrate and colonize the host tissue, which can further be utilized in the effective management of postharvest disease [36,37,38,39]. In this study, six non-pathogenic endophytic fungi including NS-3, NS-11, NS-38, NS-43, NS-56, and NS-58 appeared to demonstrate antagonistic activities against *N. parvum*. All of them were identified for the genus of *Penicillium* except for NS-56. *Penicillium* sp. has also been widely studied as a biocontrol endophytic fungus for a long time [29,40,41,42,43]. *Penicillium* sp., which was isolated from the stems of tomato, was highly effective in reducing the mycelial growth of *Fusarium oxysporum f.* sp. *Cucumerinum* [44]. The endophytic fungi *Penicillium fructuariae-cellae* had the ability to inhibit the growth of *D. sapinea* in vitro. Endophytic species such as *P. oxalicum*, *P. chrysogenum*, *P. crustosum*, *P. striatisporum*, *P. griseofulvum*, and *P. chermesinum* appeared to demonstrate remarkable biocontrol activities against plant pathogens [45]. Therefore, our findings indicate that *Penicillium* sp. from postharvest chestnuts has a significant potential as a BCA against host fungal pathogens.

It should be pointed out that the genus belonging to *Penicillium* is considered as a rich resource of bioactive metabolites [46,47,48]. There are a large number of *Penicillium* species that produce biologically active secondary metabolites, making them suitable for agricultural, biotechnological, and pharmaceutical applications [35,49,50,51,52]. The ethyl acetate active fraction of *P. chrysogenum* is mainly used for the isolation and identification of its bioactive compounds via extraction with organic solvents [53]. A number of compounds with potential biological activity have been identified in studies of *Penicillium* and its ethyl acetate extracts [54]. Furthermore, the ethyl acetate extract from the endophytic fungus *P. chrysogenum* from Liagora viscida was found to yield 12 known metabolites, including potent emodin and ω-hydroxyemodin [50]. In this study, we also found that the ethyl acetate phase extract of NS-3-38 had the strongest antifungal activity against *N. parvum*, for which its inhibitory rate was up to 90.19%. The results indicate that the ethyl acetate extract of *Penicillium* contains a diverse array of compounds, many of which exhibit notable antifungal activities against postharvest pathogens.

In the past, a number of studies have been carried out on the biocontrol effect of single antagonistic endophytic fungi on postharvest diseases of fruits and vegetables [55,56,57]. However, the single antagonistic endophytic fungi could not guarantee a stable biocontrol efficiency against a variety of pathogens, due to environmental factors. Compared with a single antagonistic endophytic fungi, antagonist mixtures have a higher activity and a better ability to adapt to environmental pressure [35]. The mixed culture of two endophytic fungal strains, *Trichoderma longibrachiatum* CSN-18 and *Aspergillus* sp. CSN-3, had a much higher inhibition rate to the mycelial growth of *P. camelliaecola* as compared to monocultures [58]. When *Aspergillus austroafricanus* was grown in mixed cultures with *B. subtilis* with *S. lividans*, several metabolites were induced up to 29-fold, which enhanced biocontrol activity against *Staphylococcus aureus* [32]. In this study, some mixed antagonistic endophytic fungi, such as NS-3-38, NS-11-38, NS-43-56, and NS-56-58-38, exhibited a much stronger antifungal activity against *N. parvum* than when applied individually. Among them, the mixture of NS-3-38 showed the highest antifungal activity, for which its inhibition rate was up to 86.67% ± 0.45%. However, there are some combinations of antagonistic endophytic fungi such as NS-3-38-43 and NS-56-58-11 that showed much weaker inhibitory effects on the pathogen compared with the single ones, which may be related to the compatibility of endophytic fungi. Furthermore, the fermentation broth mixture of *P. chermesinum* and *P. decaturense* along with chitosan coating greatly decreased the fruit decay and weight loss of chestnuts during storage. Therefore, our findings indicate that reasonable designs and applications are required when using mixed endophytic fungi, and *P. chermesinum* mixed with *P. decaturense* has a significant potential to be developed as a promising *complex* BCA for postharvest disease management in chestnut fruits.

## 5. Conclusions

In total, 6 strains of antagonistic endophytic fungi against *N. parvum* were isolated from 300 “Dahongpao” chestnut fruits. Some mixed antagonistic endophytic fungi, such as NS-3-38 (*P. chermesinum* and *P. decaturense*), exhibited a stronger antifungal activity against *N. parvum* than when applied individually. At the same time, the crude extracts from *P. chermesinum* and *P. decaturense* also had strong pathogen inhibitory effects, in which the active components lie in the ethyl acetate fraction. The culture filtrate extracts from *P. chermesinum* and *P. decaturense* combined with a chitosan coating significantly reduced the fruit decay and weight loss of chestnuts during storage. These beneficial microorganisms in the present study will accelerate the development of novel mixed biological control agents to prevent the fruit decay of postharvest chestnuts.

## Figures and Tables

**Figure 1 jof-10-00573-f001:**
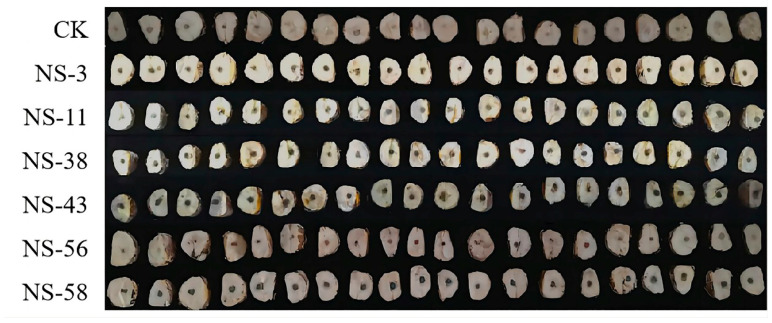
Pathogenicity assay of endophytic fungi from postharvest chestnuts using artificial inoculation. CK stands for PDA inoculation only, without endophytic fungi.

**Figure 2 jof-10-00573-f002:**
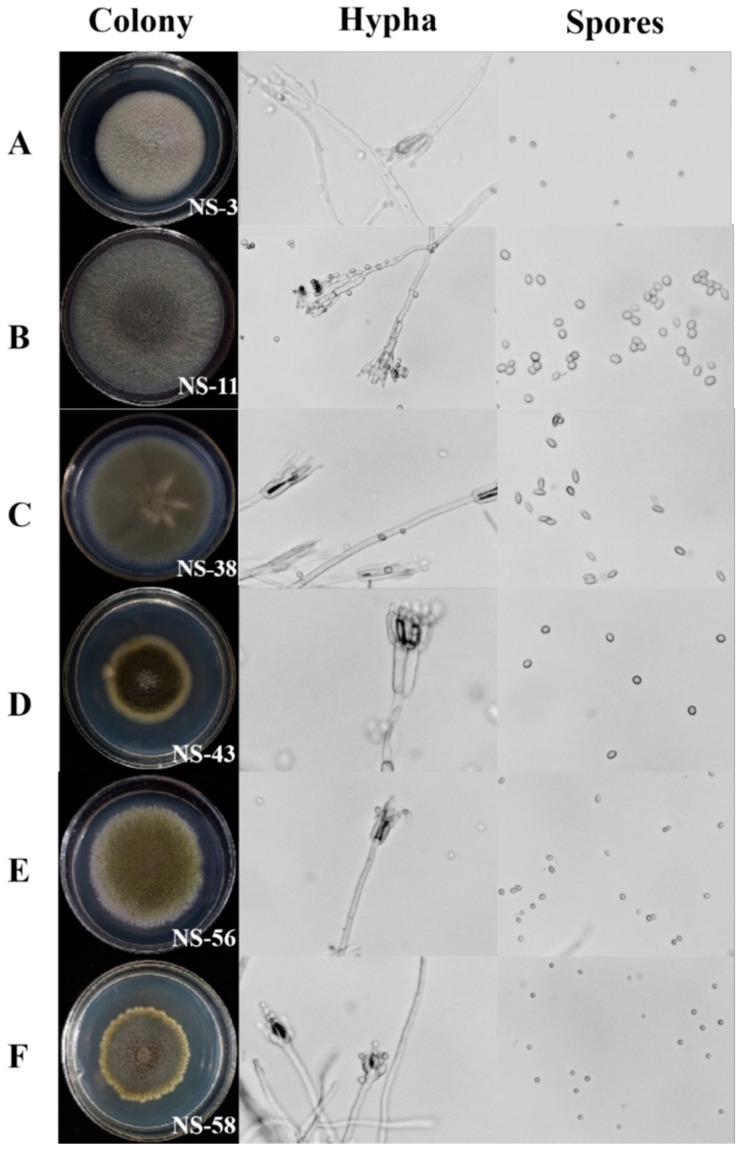
Morphology analyses of colony, hypha, and spores from endophytic antagonistic fungi from postharvest chestnuts. (**A**) NS-3; (**B**) NS-11; (**C**) NS-38; (**D**) NS-43; (**E**) NS-56; (**F**) NS-58. Microscope with 40× magnification.

**Figure 3 jof-10-00573-f003:**
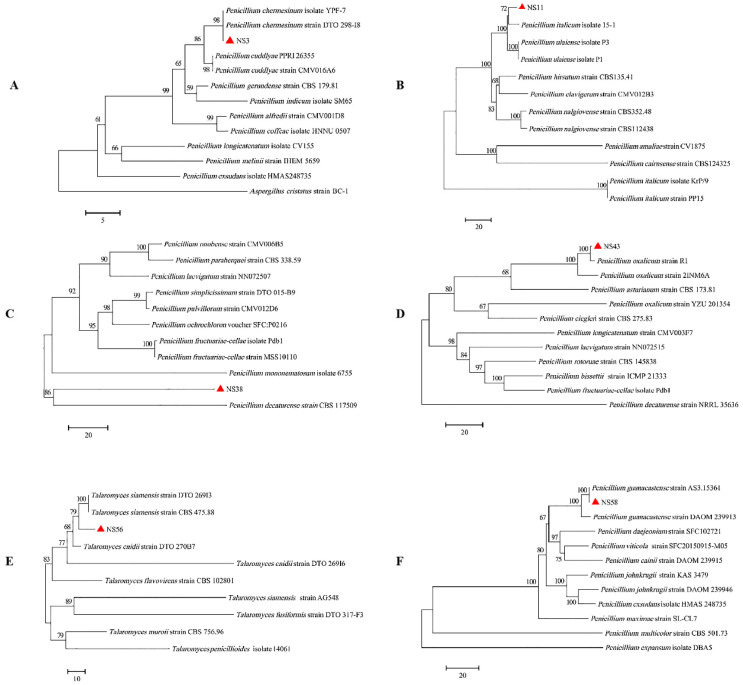
Phylogenetic tree constructed based on ITS and BenA gene sequences. (**A**) NS-3; (**B**) NS-11; (**C**) NS-38; (**D**) NS-43; (**E**) NS-56; (**F**) NS-58.

**Figure 4 jof-10-00573-f004:**
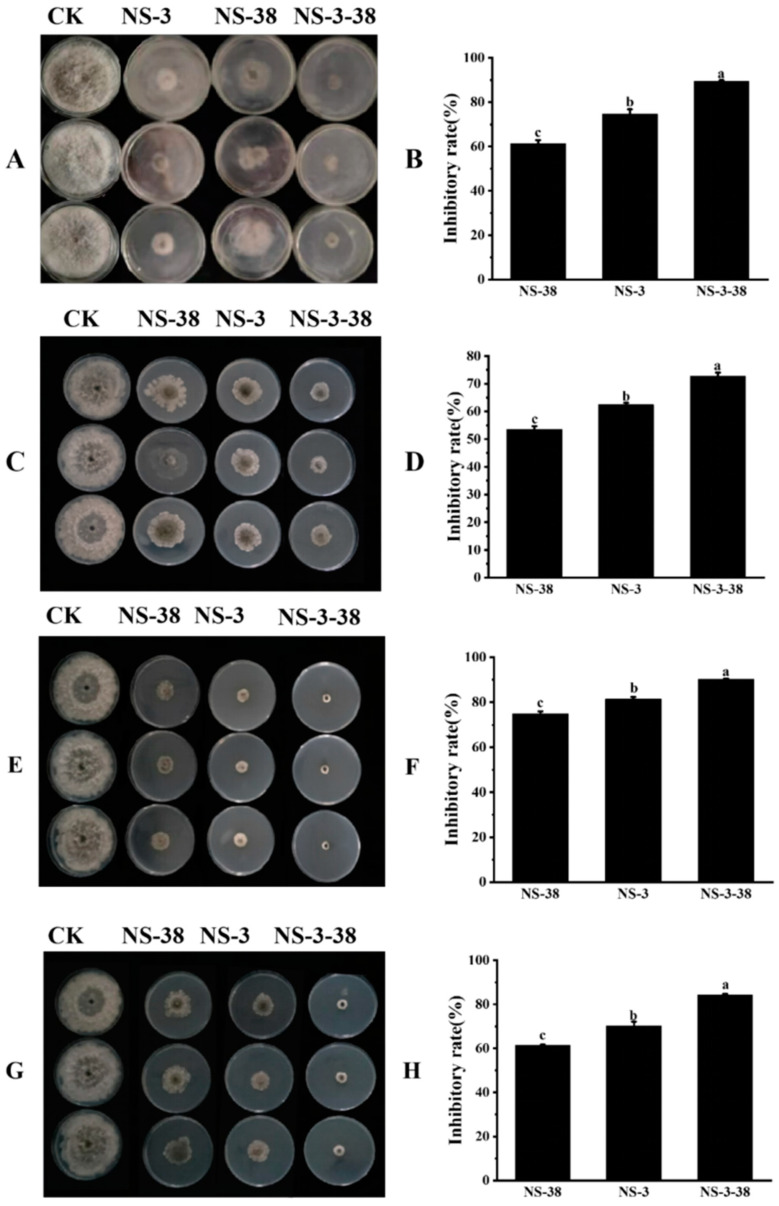
Antagonistic activity and inhibitory rate of the culture filtrate from endophytic fungi and its fractions against *N. parvum*. (**A**,**B**) culture filtrate; (**C**,**D**) fraction extracted using petroleum ether extract treatment; (**E**,**F**) fraction extracted using ethyl acetate extract treatment; (**G**,**H**) fraction extracted using 1-butanol. Different letters (a, b, c) above the columns indicate significant difference between the groups (*p* < 0.05).

**Figure 5 jof-10-00573-f005:**
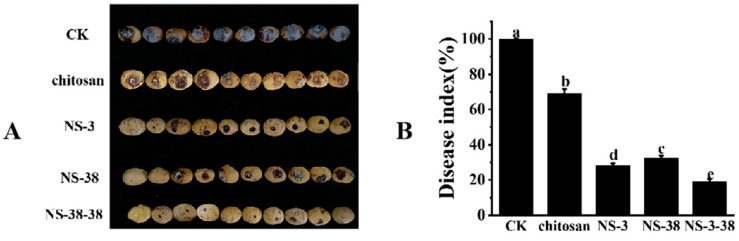
Effects of endophytic antagonistic fungi culture filtrate combined with chitosan coating on disease index of chestnut fruits caused by *N. parvum*. (**A**) Appearance of chestnut fruits inoculated with *N. parvum*. (**B**) Disease index of chestnut fruits.Different letters (a, b, c, d and e) above the columns indicate significant difference between the groups (*p* < 0.05).

**Figure 6 jof-10-00573-f006:**
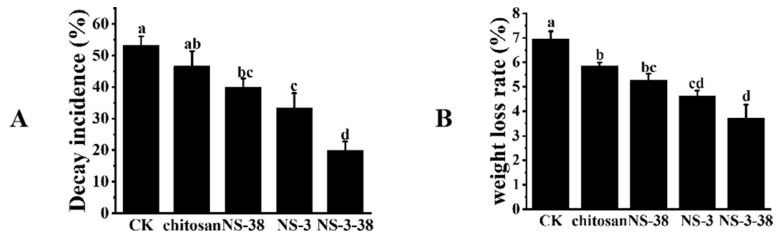
Effects of endophytic antagonistic fungi culture filtrate combined with chitosan coating on decay incidence and weight loss rate of chestnut during storage. (**A**) Decay incidence; (**B**) weight loss rate.Different letters (a, b, c, d) above the columns indicate significant difference between the groups (*p* < 0.05).

**Table 1 jof-10-00573-t001:** Gene locus and primer sequences used for molecular identification.

Genetic Locus	Primer Name	Primer Sequence (5′-3′)
BenA	BT_2_a-F	GGTAACCAAATCGGTGCTGCTT
	BT_2_a-R	ACCCTCAGTGTAGTGACCCTTGGC
CaM	CMD5-F	CCGAGTACAAGGARGCCTTC
	CMD6-R	CCGATRGAGGTCATRACGTGG
ITS	ITS1	TCCGTAGGTGAACCTGCGG
	ITS4	TCCTCCGCTTATTGATATGC

**Table 2 jof-10-00573-t002:** Inhibition rates of endophytic fungi against *N. parvum* of chestnuts.

Strains	Inhibition Rate (%)	Strains	Inhibition Rate (%)
NS-3	51.85 ± 1.39 ^k^	NS-38-43	76.67 ± 1.98 ^b^
NS-38	69.26 ± 1.72 ^cde^	NS-38-56	75.93 ± 0.26 ^b^
NS-3-38	86.67 ± 0.45 ^a^	NS-38-58	75.00 ± 2.36 ^bc^
NS-11	44.07 ± 1.14 ^lm^	NS-43-56	77.22 ± 1.64 ^b^
NS-3-11	43.15 ± 1.83 ^mn^	NS-43-58	76.11 ± 0.45 ^b^
NS-43	59.26 ± 1.59 ^fj^	NS-56-58	67.59 ± 3.28 ^de^
NS-3-43	44.81 ± 5.02 ^lm^	NS-3-38-11	55.37 ± 5.00 ^jk^
NS-56	45.19 ± 4.12 ^lm^	NS-3-38-43	36.48 ± 2.58 ^op^
NS-3-56	56.30 ± 2.77 ^jk^	NS-3-38-56	41.85 ± 3.86 ^no^
NS-58	42.22 ± 0.45 ^mno^	NS-3-38-58	65.93 ± 2.24 ^de^
NS-3-58	50.00 ± 8.03 ^kj^	NS-43-56-58	62.78 ± 1.57 ^ef^
NS-11-38	75.93 ± 3.86 ^b^	NS-56-58-11	36.85 ± 1.14
NS-11-43	76.48 ± 2.10 ^b^	NS-56-58-38	77.59 ± 2.58 ^b^
NS-11-56	74.63 ± 1.39 ^bc^	NS-56-58-43	72.41 ± 2.05 ^bcd^
NS-11-58	69.07 ± 1.14 ^cde^	NS-56-58-3	67.04 ± 0.69 ^cd^
		NS-F	32.59 ± 4.89 ^p^

NS-F represents a combination of six strains of endophytic fungi; NS-3-38 indicates a combination of NS-3 and NS-38. Different letters (a, b, c, d, e, f, j, k, l, m, n, o and p) above the columns indicate significant difference between the groups (*p* < 0.05).

## Data Availability

The original contributions presented in the study are included in the article, further inquiries can be directed to the corresponding author.

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
