# Peer review of "Isolation of Antagonistic Endophytic Fungi from Postharvest Chestnuts and Their Biocontrol on Host Fungal Pathogens"

_jof, 2024, doi:10.3390/jof10080573_

Round 1
Reviewer 1 Report
Overall, the experiment designs are fair, and the conclusions are reasonable. However, there is much missing information about the details of the experiments in the Materials and Methods section. This section needs to be improved.
A "synergistic" effect is defined as when two or more treatments are combined, their effects are more than the sum of those treatments. "1 + 1 = more than 2" is the synergistic effect. However, in this study, 1 + 1 = 1.2 or 1.3. I wouldn't call this a "synergistic effect." The authors should remove the word "synergistic" from the title and text.
L11: Spell out "N. parvum".
L12: 6 strains of what fungal species?
L14: Spell out "P" of "P. chermesinum".
L17: -> mixture of NS-3 and NS-38
L23: delete "obviously", instead use "significantly" if there was a statistical significance.
L51: Spell out "A. pullulans"
L63: -> P. griseofulvum
L73: stored for how many days or weeks? Storage condition?
L74-L77: Cite reference for ITS identification/primer used in this study.
L81: delete "finally".
L84-86: Explain how you obtained pure culture? Single spore isolation??
L90: what medium?
L106: Cite a paper for MEGA 7.0 "Kumar et al. (2016) Mol. Biol. Evol. 33:1870-1874".
L110: delete "finally"
L134-137: What is the percentage (%) of those petroleum ether etc. are in the extracted solution? Do the control contain the similar percentage of petroleum ether etc. when comparing the results??
L149-159: Explain how you used culture filtrate to this assay. How many fruit were used for this assay per treatment?
L151: What's the depth of the wound? Explain how you made a wound on chestnuts.
L153: what's 4-day-old fungal disc? How did you get it? What's the size of the disc? ...inoculated "on the wound"? ...the inoculated fruit were then stored at 26C...
L160: how many fruit were tested per treatment?
L161: Delete "selected"
L163: how did you store those fruit after air-dry? Explain storage condition of fruit.
L164: When did you take the measurement of weight loss and decay incidence during the storage? How many times?
L167: It explains the initial and final, state so in L161-165. "The weight data was taken before and after the storage and decay incidence was recorded after the storage".
L146, 160, 171: the subtitle number should be 2.8, 2.9,..
Fig. 1: Show fewer and bigger images. Images are too small to see.
Table 2: It's confusing that the strain name "NS-3-38" is a mixture of "NS-3" and "NS-38". Use different description such as "NS-3 + NS-38" or "NS-3/38" and STATE so in the text and in the footnote of the tables.
L194: "Synergistic" effect is defined as " interaction of two or more drugs when their combined effect is greater than the sum of the effects ". Usually 1+1= more than 2. In the table 2, it's almost 1 + 1 = 1.2. I wouldn't call those effects "synergistic". Remove "synergetic" from the title.
L197-200: Explain more in detail.
L205-217: You need to provide the size of conidia for those 6 strains, and compared the data with that of type species in Discussion section.
L221: NJ method is obsolete. Use Maximum likelihood or more modern method for the future reference.
Fig.3 Combine all Penicillium trees into one with more type species from the database.
L246. Fig. ?
Fig5A. Make it bigger.
L262,263: What's "disease index"? Explain it in M&M section.
L300, L310, L314: Penicillium -> P
L347: delete "much", or write otherwise "...significantly inhibited fungal growth of N. paravum than when applied individually".
Author Response
Dear editor and reviewer,
Thank you all so much for your reviewing my work (Manuscript ID:jof-3077457, Title: Isolation of antagonistic endophytic fungi from postharvest chestnut and their synergetic biocontrol on host fungal pathogen).
First, I want to show my respect to all the reviewers, because they are competent, and their comments and suggestions are highly reasonable, and help me a lot.
According to their comments, I have tried my best to revise my manuscript. And now, I will give my detailed explanations for the revision:
Editorial comments:
Reviewer 1
Comments 1:L11 Spell out "N. parvum".
Response1:
Thank you very much for the constructive suggestions, the "N. parvum" has been spelled out on Page 2, Line 26.
Comments 2:L12: 6 strains of what fungal species?
Response2:
Thank you very much for the constructive suggestions. All six strains of fungi are endophytic penicilli, the six endophytic fungal strains were Penicillium Chermesinum (NS-3), Penicillium Italicum (NS-11), Penicillium Decaturense (NS-38), Penicillium Oxalicum (NS-43), Talarmyces siamensis (NS-56), Penicillium Guanacastense (NS-58).
Comments 3:L14: Spell out "P" of "P. chermesinum".
Response3:
Thank you very much for the constructive suggestions,the "P. chermesinum" has been spelled out on Page 2, Line 2.
Comments 4:L17: -> mixture of NS-3 and NS-38
Response4:
Thank you very much for the constructive suggestions,the mixture of NS-3 and NS-38 has been changed on Page 2, Line 31.
Comments 5:L23: delete "obviously", instead use "significantly" if there was a statistical significance.
Response5:
Thank you very much for the constructive suggestions,the "obviously" has been instead use "significantly" on Page 2, Line 38.
Comments 6:L51: Spell out "A. pullulans"
Response6:
Thank you very much for the constructive suggestions, the "A. pullulans" has been spelled out on Page 2, Line 66.
Comments 7:L63: -> P. griseofulvum
Response7:
Thank you very much for the constructive suggestions, the "P. griseofulvum" has been changed on Page 2, Line 66.
Comments 8:L73: stored for how many days or weeks? Storage condition?
Response8:
Thank you very much for the constructive suggestions, The storage time period was 16 days, and storage conditions were such that the chestnuts were placed in PE preservation bags with air permeability and stored for 16 d at room temperature (25±1°C).
Comments 9:L74-L77: Cite reference for ITS identification/primer used in this study.
Response9:
Thank you very much for the constructive suggestions, references to the ITS identification/stator used in this study have been cited on Page 2, Line 110. References include mainly [31-33]
Comments 10:L81: delete "finally".
Response10:
Thank you very much for the constructive suggestions, the "finally" has been deleted on Page 5, Line 98.
Comments 11:L84-86: Explain how you obtained pure culture? Single spore isolation?
Response11:
Thank you very much for the constructive suggestions, chestnuts were sterilised twice and washed 3 times with sterile water, inoculated into PDA medium and incubated at a constant temperature of 26°C for 24 h. After 24 h of incubation, the strains were allowed to grow on the edge of the chestnut tissues of the PDA and the young mycelium growing on the edges of the colonies were picked from the PDA and transferred to a new medium to streak the culture of the single colonies, and the incubation was continued under the same conditions, repeating the previous isolation operation for 3 times, until the pure colonies were obtained, and the main picked is mycelium.
Comments 12:L90: what medium?
Response12:
Thank you very much for the constructive suggestions, the medium mainly refers to PDA medium.
Comments 13:L106: Cite a paper for MEGA 7.0 "Kumar et al. (2016) Mol. Biol. Evol. 33:1870-1874".
Response13:
Thank you very much for the constructive suggestions, has been cited a paper for MEGA 7.0 "Kumar et al. (2016) on Page 6, Line 123.
Comments 14:L110: delete "finally"
Response14:
Thank you very much for the constructive suggestions, the "finally" has been deleted on Page 6, Line 126.
Comments 15:L134-137: What is the percentage (%) of those petroleum ether etc. are in the extracted solution? Do the control contain the similar percentage of petroleum ether etc. when comparing the results?
Response15:
Thank you very much for the constructive suggestions, the three organic solvents, petroleum aldehyde, ethyl acetate and n-butanol, are only used as carriers for the extraction of the fermentation broth, and the solutions of the extraction phases are combined, depressurised and concentrated to obtain the final extracts, which are freeze-dried at low temperatures and do not contain these organic solvents, nor do they contain these organic solvents in the control group.
Comments 16:L149-159: Explain how you used culture filtrate to this assay. How many fruit were used for this assay per treatment?
Response16:
Thank you very much for the constructive suggestions, the fungal fermentation solution combined with chitosan coating solution was applied to chestnut fruits: 1.75% chitosan-based film solution was used, and the selected chestnuts were put into the endophyte 3-38 fermentation solution chitosan-coated group, the endophyte 38 fermentation solution chitosan-coated group, endophyte 3 fermentation solution chitosan-coated group, chitosan-control group, and natural-control group respectively, and the chestnuts were dipped into the corresponding solution for 5 min (the natural-control group was not treated). The chestnuts were soaked in the corresponding solution for 5 min (the natural control group was left untreated), then removed and dried, each treatment group was inoculated with 10 chestnuts in 3 parallels each. The chestnut kernels were placed in a preservation box at room temperature, and after 5 d, the shells were peeled off and the onset of the chestnut kernels was observed.
Comments 17:L151: What's the depth of the wound? Explain how you made a wound on chestnuts.
Response17:
Thank you very much for the constructive suggestions, chestnuts of uniform size and without mechanical damage on the surface were selected, and after surface disinfection with 0.2% NaCIO solution, the wound-making inoculation method was used to cut a small square hole (5 mm wide × 5 mm long) at the waist of the fruit with a razor blade.
Comments 18:L153: what's 4-day-old fungal disc? How did you get it? What's the size of the disc? ...inoculated "on the wound"? ...the inoculated fruit were then stored at 26C...
Response18:
Thank you very much for the constructive suggestions, 4 days of incubation is the pathogen Neofusicoccum parvum in this experiment, 4 days is the growth cycle of Neofusicoccum parvum in PDA medium, and after inoculation on artificially bruised chestnut fruits then the chestnuts are left for 5 days to observe the results, 5 days is the period of time the pathogen is in the chestnut pathogenicity cycle.
Comments 19:L160: how many fruit were tested per treatment?
Response19:
Thank you very much for the constructive suggestions, fifty fruits were tested in each treatment, for a total of five treatment groups, each treatment group being a parallel.
Comments 20:L161: Delete "selected"
Response 20:
Thank you very much for the constructive suggestions, "selected" has been deleted on Page 9, Line 180.
Comments 21:L163: how did you store those fruit after air-dry? Explain storage condition of fruit.
Response 21:
Thank you very much for the constructive suggestions, air-drying refers to the natural drying of the coating solution on the surface of chestnuts treated with endophytic fungal fermentation combined with chitosan-coated immersion before storage, under the condition that the treated chestnuts were placed in PE preservation bags with air permeability and stored at room temperature (25±1°C) for 16 d. The storage conditions were as follows: the treated chestnuts were put into PE preservation bags with air permeability, and stored for 16 d at room temperature (25±1°C).
Comments 22:L164: When did you take the measurement of weight loss and decay incidence during the storage? How many times?
Response 22:
Thank you very much for the constructive suggestions, decay rate at the end of the storage period, chestnuts from all treatment groups were cut open to observe the internal decay of the chestnuts, and the decay rate on day 16 was measured, one time. The weight loss rate is determined as the value at the beginning and at the end of the day, and the difference between the two is calculated.
Comments 23:L167: It explains the initial and final, state so in L161-165. "The weight data was taken before and after the storage and decay incidence was recorded after the storage".
Response 23:
Thank you very much for the constructive suggestions, decay incidence in the early stage of storage in the early 0 days is not rot, in the late storage of 16 days before the occurrence of rot, so the rate of rot only recorded in the late storage of the value. The weight loss rate needs to be determined between the weight of the chestnut at the beginning of storage and the weight of the chestnut at the end of storage.
Comments 24:L146, 160, 171: the subtitle number should be 2.8, 2.9,..
Response 24:
Thank you very much for the constructive suggestions, the subtitle number 2.8, 2.9 has been corrected on Page 9, Line 178.
Comments 25:Fig. 1: Show fewer and bigger images. Images are too small to see.
Response 25:
Thank you very much for the constructive suggestions, The image has been resized
Comments 26:Table 2: It's confusing that the strain name "NS-3-38" is a mixture of "NS-3" and "NS-38". Use different description such as "NS-3 + NS-38" or "NS-3/38" and STATE so in the text and in the footnote of the tables.
Response 26:
Thank you very much for the constructive suggestions, NS-3-38 has been labelled in the table markings.
Comments 27:L194: "Synergistic" effect is defined as " interaction of two or more drugs when their combined effect is greater than the sum of the effects ". Usually 1+1= more than 2. In the table 2, it's almost 1 + 1 = 1.2. I wouldn't call those effects "synergistic". Remove "synergetic" from the title.
Response 27:
Thank you very much for the constructive suggestions, the word synergetic has been removed from the title on Page 9, Line 180.
Comments 28:L197-200: Explain more in detail.
Response 28:
Thank you very much for the constructive suggestions, a detailed explanation has been given.
Comments 29:L205-217: You need to provide the size of conidia for those 6 strains, and compared the data with that of type species in Discussion section.
Response 29:
Thank you very much for the constructive suggestions, the size of conidia for those 6 strains has been given on Page 31, Line 565.
Comments 30:L221: NJ method is obsolete. Use Maximum likelihood or more modern method for the future reference.
Response 30:
Thank you very much for the constructive suggestions, In the future we will adopt the use of maximum likelihood or more modern methods as a reference for.
Comments 31:Fig.3 Combine all Penicillium trees into one with more type species from the database.
Response 31:
Thank you very much for the constructive suggestions, this has been the merger of all Penicillium spp. with more type species in the database into one.
Comments 32:L246. Fig. ?
Response 32:
Thank you very much for the constructive suggestions, this has been corrected on Page 13, Line 261.
Comments 33:Fig5A. Make it bigger.
Response 33:
Thank you very much for the constructive suggestions, the Fig5A has been made bigger.
Comments 34:L262,263: What's "disease index"? Explain it in M&M section.
Response 34:
Thank you very much for the constructive suggestions, the disease index is also known as the incidence index and the infection index. It is a method of expressing the degree of morbidity, given the explanation on Page 9, Line 167-177.
Comments 35:L300, L310, L314: Penicillium -> P
Response 34:
Thank you very much for the constructive suggestions, It has been changed on Page 15, Line 315,319.
Comments 36:L347: delete "much", or write otherwise "...significantly inhibited fungal growth of N. paravum than when applied individually".
Response 36:
Thank you very much for the constructive suggestions, much have been deleted on Page 18, Line 353.
Reviewer 2 Report
The article presented by the authors is very good, however, an English edition is necessary. In addition, I suggest that an objective description of the images, graphs and phylogenetic trees be made. Thank you.
47 Please remove the period in the scientific name
48 Change the word “acticity” to activity
91 Please remove the second "and"
131 Could you explain this paragraph, the procedure that was followed is not clear. Did you incubate the sterile fermentation broth?? or did you incubate the mycelium?? Please explain in detail.
136 Please indicate whether there are 3 or 4 fractions, only 3 are mentioned: PA, EA and BE.
185 I suggest that authors include the meaning of the letters CK in the image text. It is also important to point out why there are 20 different samples for each isolate, What does each one indicate? Incubation time?
216 Please include the images at the same size
223 The use of CaM for molecular identification was not described in the methodology. I suggest including it.
223 The identity percentages shown in some of the isolates are very low (87%) and are therefore not sufficient to identify an organism at the species level. Please explain.
225 Please include the period in P. decaturense correctly
231 Please include the period in T.siamensis correctly
242 Image A needs editing, please include the image where the complete Petri dishes can be seen
249 The 4 C-H images show greater inhibition than that observed in filtered cultures of the fungus. Please discuss and/or verify that the inhibition effect is not due to the solvents used.
261 Panel A does not indicate whether there are several nuts per replicate or the same one over the days. Please explain.
Author Response
Dear editor and reviewer,
Thank you all so much for your reviewing my work (Manuscript ID:jof-3077457, Title: Isolation of antagonistic endophytic fungi from postharvest chestnut and their synergetic biocontrol on host fungal pathogen).
First, I want to show my respect to all the reviewers, because they are competent, and their comments and suggestions are highly reasonable, and help me a lot.
According to their comments, I have tried my best to revise my manuscript. And now, I will give my detailed explanations for the revision:
Editorial comments:
Reviewer 2
Comments 1: 47 Please remove the period in the scientific name
Response1:
Thank you very much for the constructive suggestions,the period in the scientific name has been removed on Page 3, Line 62.
Comments 2: 48 Change the word “acticity” to activity
Response 2:
Thank you very much for the constructive suggestions. the word “acticity” has been Changed to activity on Page 3, Page 3, Line 63.
Comments 3: 91 Please remove the second "and".
Response 3:
Thank you very much for the constructive suggestions. The second word “and” has been removed on Page 6, Line 108.
Comments 4: 131 Could you explain this paragraph, the procedure that was followed is not clear. Did you incubate the sterile fermentation broth?? or did you incubate the mycelium?? Please explain in detail.
Response 4:
Thank you very much for the constructive suggestions. Fermentation broth obtained by co-culture of mycelium and PDB medium. The endophytic strain was activated and cultured in PDA medium for 5 d. Six pieces of 5 mm bacterial cake were taken under aseptic environment and inoculated in conical flasks containing 300 mL of PDB medium and cultured for 7 d at 28°C with shaking at 120 r/min to obtain the fermentation products of the strain. The fermentation broth obtained by the above method was first filtered through sterile gauze to perform the initial separation of the bacteria from the fermentation broth, and the fermentation broth was then filtered and decontaminated to obtain the sterile fermentation broth on Page 7.
Comments 5: 136 Please indicate whether there are 3 or 4 fractions, only 3 are mentioned: PA, EA and BE.
Response 5:
Thank you very much for the constructive suggestions. Three organic phase extracts have been indicated on Page 7, Line 143
Comments 6: 185 I suggest that authors include the meaning of the letters CK in the image text. It is also important to point out why there are 20 different samples for each isolate, What does each one indicate? Incubation time?
Response 6:
Thank you very much for the constructive suggestions.The meaning of the letter CK have been marked. The 20 samples were treated to reduce variability between chestnut fruits, and the treatment conditions were the same for all 20 samples, with the aim of verifying whether the endophytic fungus was pathogenic to chestnut fruits or not on Page 29, Line 562.
Comments 7: 216 Please include the images at the same size
Response 7:
Thank you very much for the constructive suggestions.The same size image has been re-provided on Page 31, Line 564.
Comments 8: 223 The use of CaM for molecular identification was not described in the methodology. I suggest including it.
Response 8:
Thank you very much for the constructive suggestions. CaM nucleotide sequences for molecular identification methods have been added and included on Page 6, Line 112.Page 6 tabie1
Comments 9: 223 The identity percentages shown in some of the isolates are very low and are therefore not sufficient to identify an organism at the species level. Please explain.
Response 9:
Thank you very much for the constructive suggestions, In addition to the biological identification of the strain, we also combined microscopic observation with the apparent shape of the mycelial growth.
Comments 10: 225 Please include the period in P. decaturense correctly
Response 10:
Thank you very much for the constructive suggestions.The period for P. decaturense has been corrected and filled in correctly on Page 12, Line 243.
Comments 11:231 Please include the period in T.siamensis correctly
Response 11:
Thank you very much for the constructive suggestions.The period for T.siamensis has been corrected and filled in correctly on Page 12, Line 250.
Comments 12:242 Image A needs editing, please include the image where the complete Petri dishes can be seen
Response 12:
Thank you very much for the constructive suggestions.Image A has been edited to include an image where the complete petri dish can be seen. on Page 33, Line 569.
Comments 13:249 The 4 C-H images show greater inhibition than that observed in filtered cultures of the fungus. Please discuss and/or verify that the inhibition effect is not due to the solvents used.
Response 13:
Thank you very much for the constructive suggestions. Petroleum aldehydes, ethyl acetate and n-butanol were only used as carriers for the extraction of the fermentation broth, and the solutions from the extraction phases were combined and concentrated under reduced pressure to obtain the final extractive phase extracts of each extractive phase, which were ultimately formulated to be used at a certain concentration independent of the extraction phases after concentration under reduced pressure..
Comments 14:261 Panel A does not indicate whether there are several nuts per replicate or the same one over the days. Please explain.
Response 14:
Thank you very much for the constructive suggestions.Each of the above treatment groups was set up with three replications of 30 chestnuts each .
Round 2
Reviewer 1 Report
No major comments.
there is still the word "synergistic" 4 times in the manuscript. Please revise.
L23: N. parvum
L27-28: P. italicum, P. decaturense, P. oxalicum....P. guanacastense
Author Response
Dear editor and reviewer,
Thank you all so much for your reviewing my work (Manuscript ID:jof-3077457, Title: Isolation of antagonistic endophytic fungi from postharvest chestnut and their synergetic biocontrol on host fungal pathogen).
First, I want to show my respect to all the reviewers, because they are competent, and their comments and suggestions are highly reasonable, and help me a lot.
According to their comments, I have tried my best to revise my manuscript. And now, I will give my detailed explanations for the revision:
Editorial comments:
Reviewer 2
Comments 1: there is still the word "synergistic" 4 times in the manuscript. Please revise.
Response1:
Thank you very much for the constructive suggestions, the word synergistic has been changed on Page 2 and 10, Line 23,42,83,210.
Comments 2: L23: N. parvum.
Response2:
Thank you very much for the constructive suggestions, the "N. parvum" has been spelled out on Page 2, Line 24.
Comments 3:L27-28: P. italicum, P. decaturense, P. oxalicum....P. guanacastense
Response 3:
Thank you very much for the constructive suggestions, theP. italicum, P. decaturense, P. oxalicum....P. guanacastense has been spelled out on Page 2, Line 28.P. italicum, P. decaturense, P. oxalicum.... ...P. guanacastense, which you suggested in your first revision must be spelled out with a P, and which, in its first appearance, has been spelled out in accordance with your revision.